# Effect of $B_2O_3$ on the Crystallization Behavior of $CaF_2$-Based Slag for Electroslag Remelting 9CrMoCoB Steel

**Leizhen Peng [1] , Zhouhua Jiang [1,2,*], Xin Geng [1], Fubin Liu [1] and Huabing Li [1,2]**

[1] School of Metallurgy, Northeastern University, Shenyang 110819, China; neuwindqishi@163.com (L.P.); gengx@smm.neu.edu.cn (X.G.); liufb@smm.neu.edu.cn (F.L.); lihb@smm.neu.edu.cn (H.L.)
[2] State Key Laboratory of Rolling and Automation, Northeastern University, Shenyang 110819, China
* Correspondence: jiangzh63@163.com; Tel.: +86-24-8369-1689

**Abstract:** The non-isothermal crystallization characteristics of the electroslag remelting (ESR)-type slag with varied $B_2O_3$ contents were investigated by non-isothermal differential scanning calorimetry (DSC), field emission scanning electron microscopy (SEM-EDS), and X-ray diffraction (XRD). The crystallization mechanism of the $B_2O_3$-bearing slag was also identified based on kinetics analysis. The results showed that the primary crystalline phase was $CaF_2$, there was no change in the type of the primary crystal as $B_2O_3$ content increased, and the morphology of the $CaF_2$ crystal was mainly dendritic. The sequence of crystal precipitation during the cooling process was $CaF_2$ to $Ca_{12}Al_{14}O_{32}F_2$ and $MgO/MgAl_2O_4$, followed by $Ca_3B_2O_6$. The activation energy of $CaF_2$ crystallization increased firstly, then decreased and reached stability, while the activation energy of $Ca_3B_2O_6$ crystallization increased continuously with the increasing $B_2O_3$ content. The crystallization behavior of $CaF_2$ was three-dimensional growth with a constant nucleation rate. The proper $B_2O_3$ content added into the $CaF_2$-based ESR slag should be around 1.0% to limit the precipitation of the $CaF_2$ crystal to attain good surface ingot quality and stable ESR operation.

**Keywords:** ESR; $CaF_2$; slag; $B_2O_3$; crystallization

## 1. Introduction

9CrMoCoB (COST-FB2) steel was developed in the framework of the European COST (Cooperation in Science & Technology) program for the production of large-scale rotor forging with good creep property at 625 °C and 30 MPa pressure. Under these ultra-supercritical conditions, the efficiency of thermal power plants can increase by about 5%, and $CO_2$ emissions can reduce by about 10% compared with the world average level at present [1–3]. The rotor is one of the most important components of steam turbines and its working condition is severe. Given that, high metallurgical quality is required. Most turbine manufacturers employ the electroslag remelting (ESR) process to produce the rotor ingot. ESR is a secondary refining technique that is used for the production of some varieties of high-grade specialty steels and alloys due to its extraordinary advantages [4,5]. During the molten steel droplet formation and dropping through the molten slag pool process, the good contact between the molten slag and steel makes the steel refined by removing the harmful elements and nonmetallic inclusions or reacting with the molten slag to adjust the compositions of the steel. During the ESR remelting 9CrMoCoB process, the chemically active and main reinforcing element boron content is hard to be controlled within the target range, leading to the unqualified ingot. It is an effective countermeasure to inhibit the loss of B by adding $B_2O_3$ into the molten slag by decreasing the degree of slag–metal reactions [6,7]. However, the surface quality of the ingot is another important focus during ESR

production, and the surface quality of the ESR ingot is related to the molten slag [8]; generally, it is a consequence of inappropriate lubrication and horizontal heat transfer in mold through the slag film, besides the melting rate and electrode immersion depth [9,10]. Horizontal heat transfer of the slag is strongly dependent on the crystallization characteristics of the slag [11,12]. Therefore, it is necessary to study the crystallization behavior of the ESR-type $B_2O_3$-bearing slag to control the appropriate horizontal heat transfer across the slag film in mold during the ESR process.

Previous researches have demonstrated that a small amount of $B_2O_3$ addition into the molten slag has a significant effect on the crystallization behavior of the metallurgical slag. Yan [13] reported that the $B_2O_3$ addition inhibited the crystallization of the low fluoride $CaO–Al_2O_3$ mold flux. Park [14] showed that $B_2O_3$ was beneficial for the $MgAl_2O_4$ crystal to precipitate in the $CaO–SiO_2–Al_2O_3–MgO$ based slag. Wei [15] reported that $B_2O_3$ increased the crystallization incubation time and decreased the crystallization temperature of the low fluoride mold flux. However, the previous extensive studies were focused on the effect of $B_2O_3$ on the crystallization behavior of the free-fluoride and low-fluoride mold fluxes for the steel continuous casting process. The role of $B_2O_3$ additions on the crystallization behavior of ESR type $CaF_2$-based slag has not been reported yet.

In order to develop a specific slag for ESR remelting the B microalloyed heat resistance steel, the influence of $B_2O_3$ on the crystallization behavior of ESR slags was investigated using differential scanning calorimeter (DSC) analysis in the present work. The microstructure and crystalline phases in solidified slag were determined by scanning electron microscope (SEM) equipped with energy dispersive X-ray spectroscopy (EDS) and X-ray diffraction (XRD). The crystalline phase formation of the molten slags was also calculated by Factsage 7.2, and the calculated results were compared with the experimental results. The non-isothermal melt crystallization kinetics were also investigated.

## 2. Experiments

### 2.1. Slag Sample Preparation

The slag (slag compositions are shown in Table 1) was premelted using the reagent-grade powders of *wt* (CaO) ≥ 97%, *wt* (MgO) ≥ 98%, *wt* ($CaF_2$) ≥98.5%, *wt* ($Al_2O_3$) ≥ 98.5%, and *wt* ($B_2O_3$) ≥ 98%. The thoroughly mixed powders were premelted at 1773 K (1550 °C) in a graphic crucible lined with a 0.2-mm thick molybdenum film at the even temperature zone of the $MoSi_2$ furnace shown in Figure 1. For composition homogeneity, the slags were held at 1773 K for 50 min and then furnace-cooled to room temperature.

The crushed premelted slags were subjected to chemical analysis. The $B_2O_3$ and $SiO_2$ contents were analyzed in the national analysis center for iron and steel (CISRI) using the inductively coupled plasma atomic emission spectroscopy (ICP-AES) method, and the other main chemical compositions of the slags were analyzed using the X-ray fluorescence spectroscopy (Rigaku ZSX PrimusII, Akishima, Tokyo, Japan). The chemical compositions of the premelted slag are also shown in Table 1.

**Table 1.** The compositions of the slag/wt/%.

| Slags number | Before Premelted | | | | | After Premelted | | | | | |
|:---:|:---:|:---:|:---:|:---:|:---:|:---:|:---:|:---:|:---:|:---:|:---:|
| | $CaF_2$ | CaO | $Al_2O_3$ | MgO | $B_2O_3$ | $CaF_2$ | CaO | $Al_2O_3$ | MgO | $B_2O_3$ | $SiO_2$ |
| #10 | 55 | 20 | 22 | 3.0 | 0.0 | 53.10 | 21.32 | 21.83 | 3.10 | 0.00 | 0.15 |
| #11 | 55 | 20 | 22 | 3.0 | 0.5 | 52.90 | 21.15 | 21.71 | 3.08 | 0.47 | 0.18 |
| #12 | 55 | 20 | 22 | 3.0 | 1.0 | 52.68 | 21.02 | 21.62 | 3.07 | 0.97 | 0.16 |
| #13 | 55 | 20 | 22 | 3.0 | 1.5 | 52.42 | 20.92 | 21.54 | 3.05 | 1.45 | 0.18 |
| #14 | 55 | 20 | 22 | 3.0 | 2.0 | 52.18 | 20.79 | 21.43 | 3.03 | 1.94 | 0.15 |
| #15 | 55 | 20 | 22 | 3.0 | 3.0 | 51.69 | 20.59 | 21.23 | 3.02 | 2.89 | 0.18 |

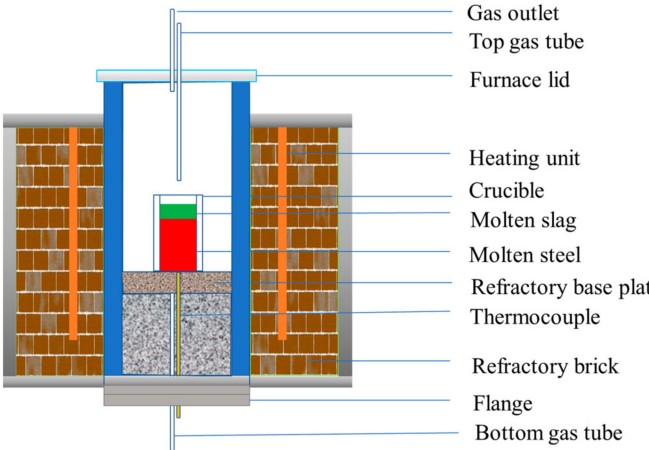

**Figure 1.** Schematic of experimental resistance furnace.

## 2.2. DSC Measurement

The crystallization behaviors of crushed premelted slags were carried out in a power compensation NETZSCH Instrument DSC (STA409CD, NETZSCH-Gerätebau GmbH, Selb, Germany). Temperature calibration and enthalpy calibration were performed before measurements using high-purity metals of In, Zn, Al, and Au with known melting points and enthalpies. Then, the sample was placed in a platinum crucible of 50 mL and an empty crucible of the same type was used as a reference in an Ar-purged chamber (Ar purity > 99.999%) at a flow rate of 50 mL·min$^{-1}$. Approximately 8 mg of pulverized premelted slag was heated up at a constant heating rate of 10 K·min$^{-1}$ from room temperature to 1773 K (1500 °C) and then held for 5 min to homogenize its chemical composition at 1773 K. Subsequently, the liquid slag sample was cooled at a constant cooling rate of 10 K·min$^{-1}$ to 623 K (350 °C). Furthermore, on account of the buoyancy or the impact of the convection and turbulence of the gas, the baselines have been employed to calibrate the DSC curves.

## 2.3. Analysis of the Crystalline Phases

The macrochanges (the DSC curves of the six slags) are often related to the microstructure changing [16,17] (the crystalline phase and its mophology). Thus, after the DSC masurements, the microstrures should be analyzed.

The samples after DSC measurements are too small to determine the crystalline phase by XRD. Hence, after DSC measurements, a series of continuous cooling experiments were carried out in order to determine the precipitated crystalline phases of the slag samples based on the corresponding temperature of each exothermic peak on the DSC curves.

The water-quenching method is a way to solidify the structure of melting slag at high temperature and then study the quenching slag at room temperature, which is common in the precipitated phase investigation. During the water-quenching experiment process, the graphite crucible (10 mm in inner diameter, 20 mm in height) was used. The crucible was hung up with the molybdenum wire, and approximately 1 g of the crushed premelted slag was heated to 1773 K in the crucible at the even temperature zone of the MoSi$_2$ furnace shown in Figure 1, and held for 5 min to achieve better homogeneity. Then, the molten slag samples were cooled at about 10 K·min$^{-1}$ to the corresponding temperature of each exothermic peak on the DSC curves. Afterward, the graphite crucible containing the molten slag was pulled out and put swiftly into the iced water. The crystalline phases of the quenched slags were identified by XRD. The XRD measurements were carried out on the Ultima IV (1.6 kW) X-ray diffractometer equipped with graphite crystal monochromator and Cu–K$_\alpha$ radiation in a 2θ ranging from 15 to 90 degrees with a scanning rate of 1.25 deg·min$^{-1}$. The crystalline phases of the quenched slags were identified by ZEISS ULTRA PLUS field-emission scanning electron microscope

equipped with energy dispersive X-ray spectroscopy (FESEM-EDS), and the operating voltage was 20 kV.

### 2.4. Thermodynamic Equilibrium Calculations

The thermodynamic analysis software package Factsage 7.2 was used to make predictions of multiphases equilibria and the proportion of solid phases for a multicomponent system [18,19]. In this study, the equilibrium phases from 1400 °C to 900 °C with a 20 °C interval were calculated. The FToxid and FactPS databases were selected during the calculation. The results were used as a reference to interpret the crystallization change and compared with the quenching products.

### 2.5. Activation Energy of Crystallization

The activation energies of crystallization were determined by the Freeman–Carroll method [20–23]. The Freeman–Carroll method is one of the most commonly methods to analyze the crystallization kinetics based on DSC/DTA experiments. It has been satisfactorily applied to describe the crystallization of metallurgical slags [24–26]. The theoretical basic for interpreting DSC/DTA results is provided by the formal theory of transformation kinetics as developed by Borchardt and Daniels [27]. The evolution of the volume fraction crystallized $\alpha$ is described as Equation (1) [27].

$$\frac{d\alpha}{dT} = \frac{A}{\beta} \times e^{-\frac{E}{RT} \times (1-\alpha)^n} \tag{1}$$

Taking one logarithm, Equation (1) yields:

$$\ln \frac{d\alpha}{dT} = \ln \frac{A}{\beta} + n \times \ln(1-\alpha) - \frac{E}{RT} \tag{2}$$

$$\frac{\Delta \lg\left(\frac{d\alpha}{dT}\right)}{\Delta \lg(1-\alpha)} = n - \frac{E}{2.3 \times R} \times \frac{\Delta\left(\frac{1}{T}\right)}{\Delta \lg(1-\alpha)} \tag{3}$$

where the $\alpha$ is the volume fraction crystallized or degree of crystallization; $E$ is the activation energy; $R$ is the gas constant; $T$ is the Kelvin temperature; $\beta$ is the cooling rate; $n$ is the Avrami parameter, which depends on the mechanism of growth and the dimensionality of the crystal; and A is the constant.

Plotting $\Delta\lg(d\alpha/dT)/\Delta\lg(1-\alpha)$ versus $\Delta(1/T)/\Delta\lg(1-\alpha)$ gives the value of $E$ and the value of $n$.

## 3. Results and Discussion

### 3.1. The Results of DSC Analysis

The effect of $B_2O_3$ addition on the DSC cooling curves of the slag samples at a cooling rate of 10 K·min$^{-1}$ are shown in Figure 2a. The exothermic peak on the DSC curve is an indication of crystalline phase precipitation [28]. As shown in Figure 2a, only two exothermic peaks were found on the DSC cooling curve for the #10 slag and three exothermic peaks for the #11–15 slags, indicating the presence of two successive crystallization events for the #10 slag, and three successive crystallization events for the #11–15 slags during a continuous cooling process. The exothermic peaks on DSC curves were named P1, P2, and P3, respectively, according to the crystalline phase formation order in the continuous cooling process.

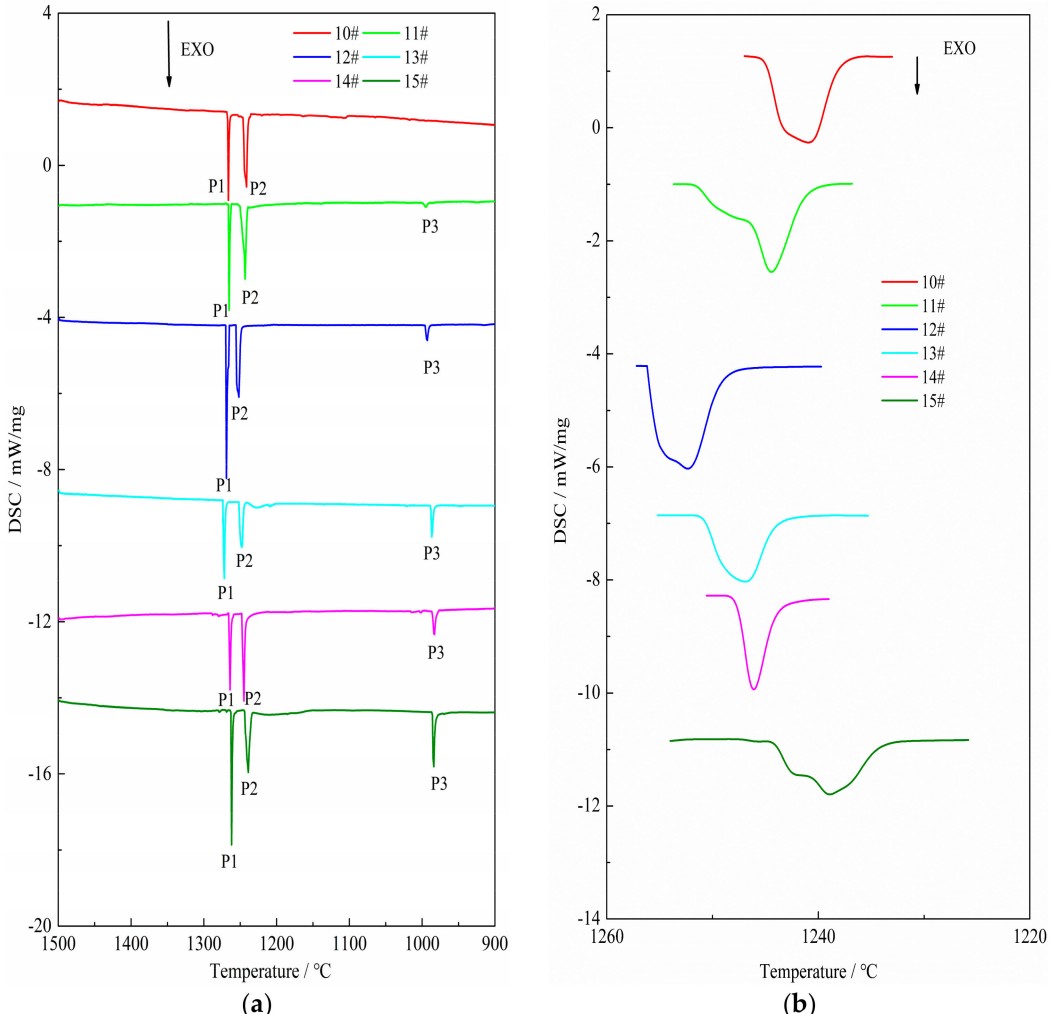

**Figure 2.** (**a**) Results of differential scanning calorimetry (DSC) analysis of six heats at rate of 10 K/min during cooling process, (**b**) the enlarged of the exothermic peak named P2.

In Figure 2a, the exothermic peak named P3 became sharper with the increasing $B_2O_3$ addition, while a sharp exothermic peak signifies a higher crystallizability [29]. The crystallization temperature of the crystalline phases named P1 and P2 fluctuated slightly, while the crystallization temperature of the crystalline phase named P3 decreased continuously with the $B_2O_3$ addition increasing.

The detailed view of the exothermic peak named P2 of the #10–15 slags are shown in Figure 2b. The exothermic peak named P2 was an overlaying peak, especially for the #10–12 and #15 slags. In the #13 and #14 slags, the crystalline phases may precipitate simultaneously. To identify the detailed crystalline phases during the cooling process, XRD analysis was needed.

### 3.2. XRD Identification

The XRD patterns of the slag samples quenched at the different temperatures are presented in Figure 3.

In Figure 3, the first exothermic peak named P1 represents the formation of $CaF_2$ in all the six slags. The second exothermic peak represents the formation of $Ca_{12}Al_{14}O_{32}F_2$ and MgO in the #10–12 slags, while in the #13 slag, the second exothermic peak represents the formation of $Ca_{12}Al_{14}O_{32}F_2$, MgO, and $MgAl_2O_4$. In the #14 and #15 slags, the second exothermic peak represents the formation of $Ca_{12}Al_{14}O_{32}F_2$ and $MgAl_2O_4$. In the #15 slag, the third exothermic peak named P3 was identified as $Ca_3B_2O_6$. However, the $Ca_3B_2O_6$ crystalline phase in the #11–14 slags cannot be identified by XRD analysis. This may be due to the slag around 980 °C being too viscous to inhibit the precipitation of

$Ca_3B_2O_6$ and the growth of the crystal. The crystalline phase amount of $Ca_3B_2O_6$ in the #11–14 slags was too small to be identified by XRD analysis.

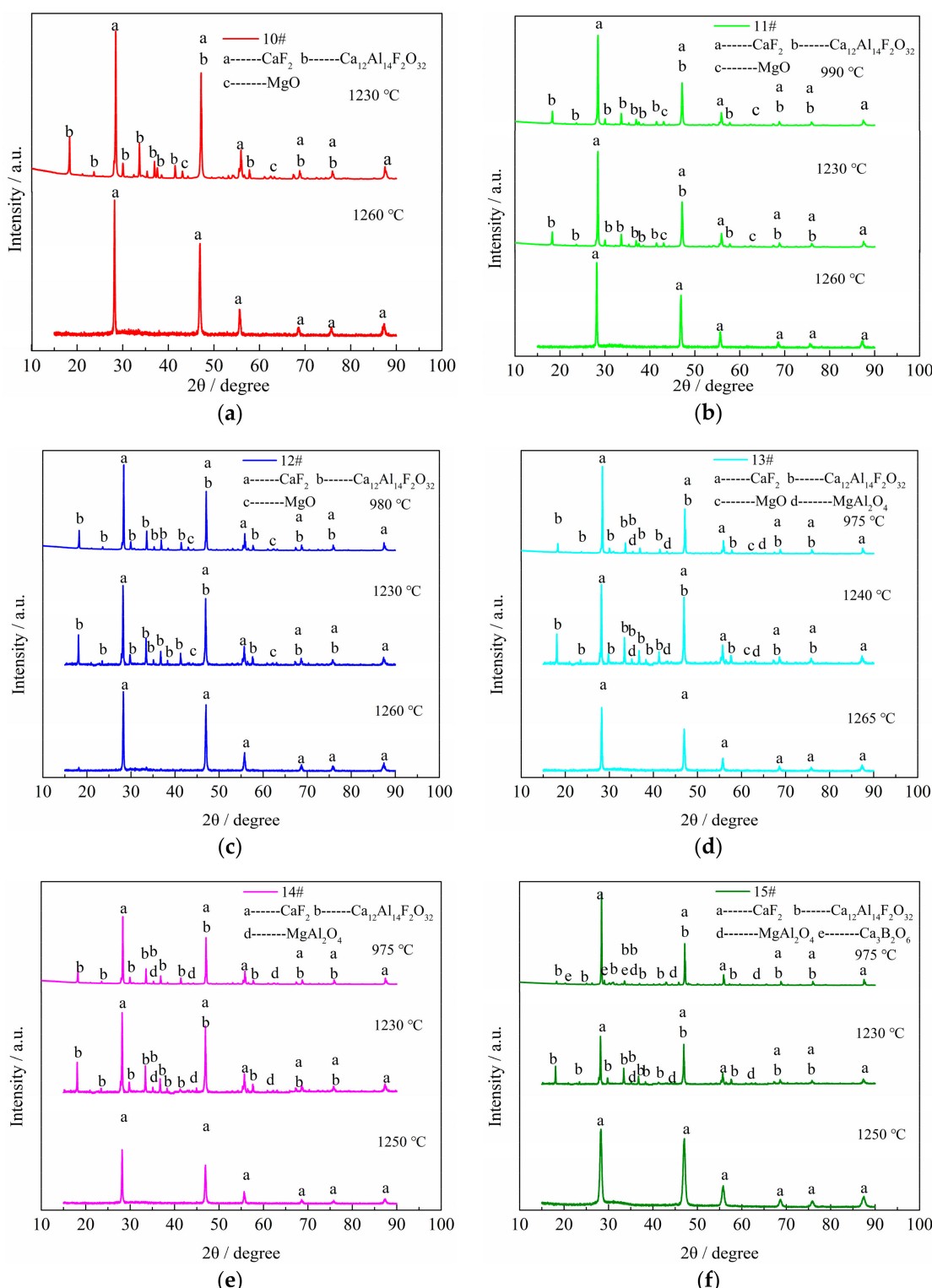

**Figure 3.** XRD patterns of the slags quenched at the desired temperatures: (**a**) #10 slag, (**b**) #11 slag, (**c**) #12 slag, (**d**) #13 slag, (**e**) #14 slag, (**f**) #15 slag.

### 3.3. SEM-EDS Analysis

The morphologies and compositions of crystalline phases in the solidified slags quenched at different temperatures were determined by SEM-EDS. The SEM back-scattered electron (BSE) images of #10–15 slags quenched at the temperature of the exothermic peak named P1 were shown in Figure 4.

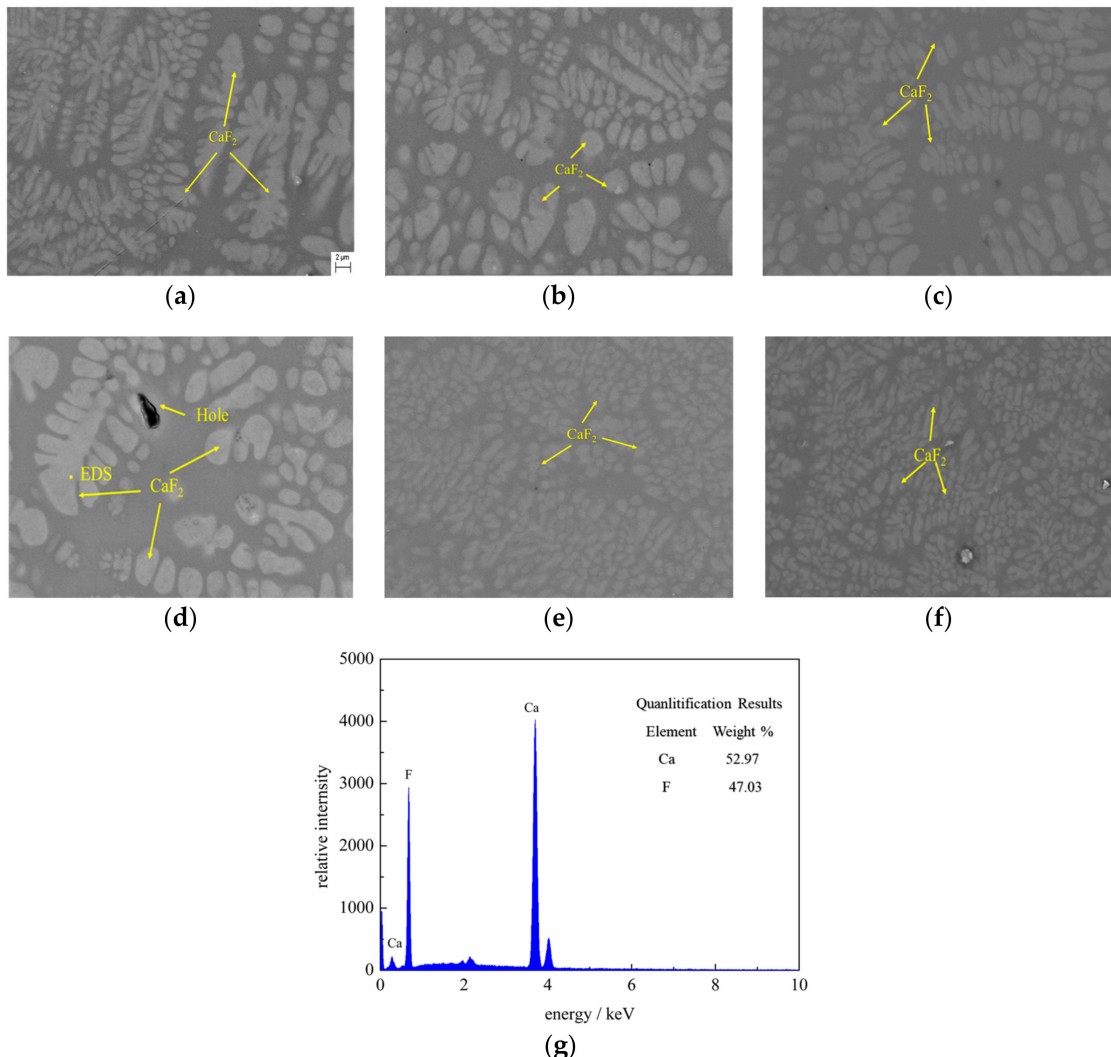

**Figure 4.** Back-scattered electron (BSE) images of the #10–15 slags quenched at the desired temperature of the exothermic peak named P1: (**a**) #10 slag, (**b**) #11 slag, (**c**) #12 slag, (**d**) #13 slag, (**e**) #14 slag, (**f**) #15 slag, and (**g**) the electroslag remelting (EDS) analysis of the crystalline phase.

In Figure 4g, the precipitated crystalline phase was identified as $CaF_2$. The morphology of the crystalline phase of $CaF_2$ in Figure 4a–f was mainly dendritic.

The SEM-EDS results of the #10 slag quenched at the temperature of the exothermic peak named P2 are shown in Figure 5.

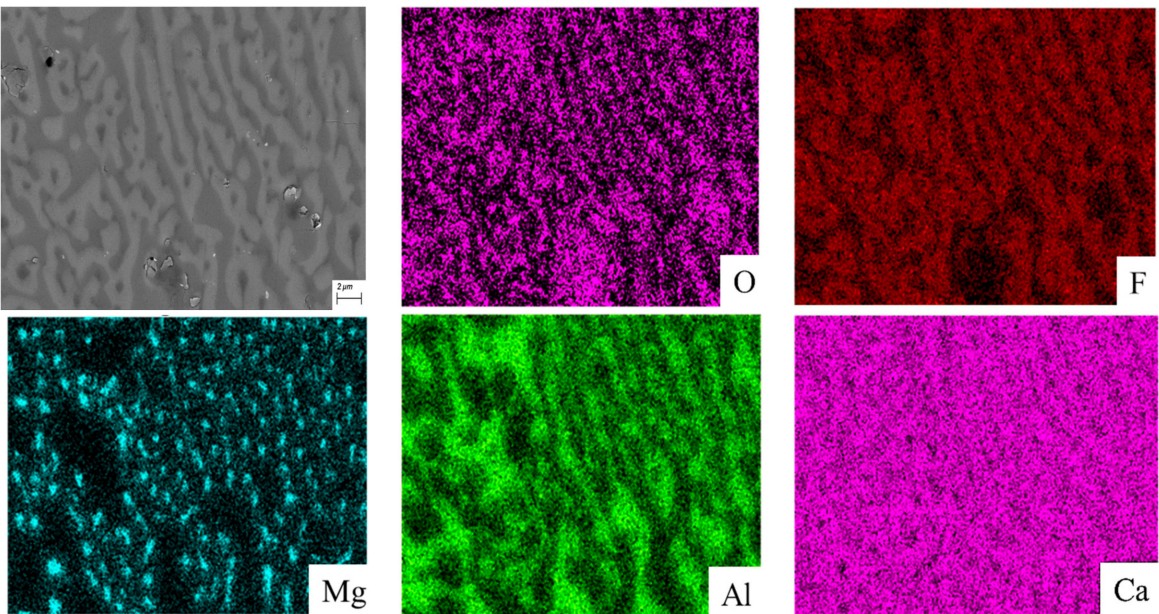

**Figure 5.** Element mappings of crystals in the #10 slag quenched at the temperature of the exothermic peak named P2.

In Figure 5, the $CaF_2$ crystal was the dominant crystalline phase and occupied the largest crystalline fraction. The MgO crystal was dispersed with the faceted morphology. The crystalline phases found in the #10 quenched slag were consistent with the XRD analysis.

The SEM-EDS results of the #11 slag quenched at the temperature of the P3 exothermic peak are shown in Figure 6, and the results of the #12 slag are shown in Figure 7.

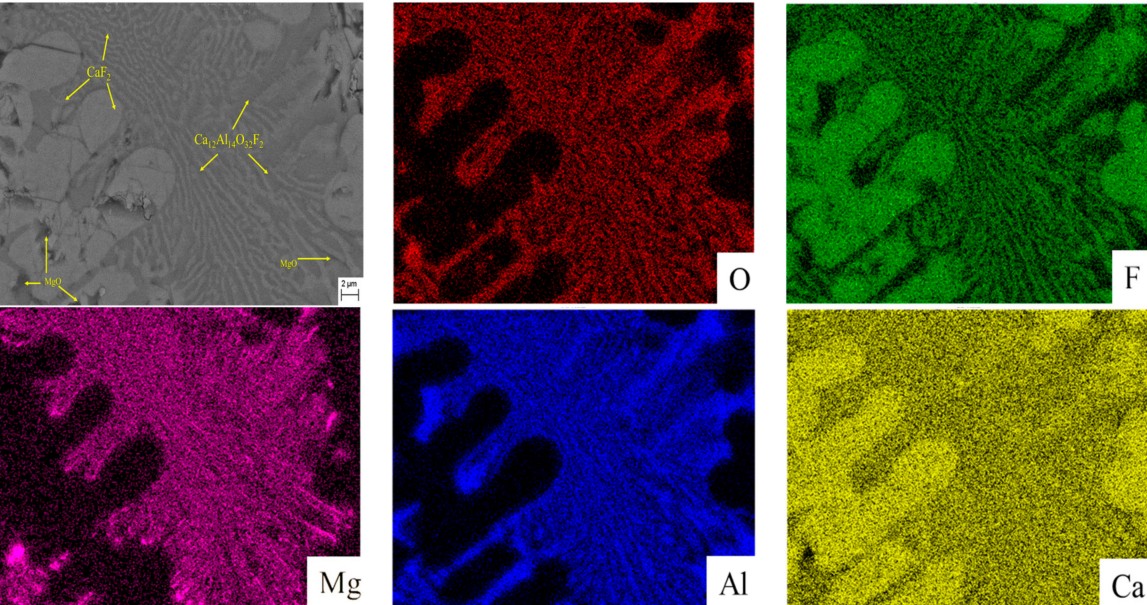

**Figure 6.** Element mappings of the #11 slag quenched at the temperature of the exothermic peak named P3.

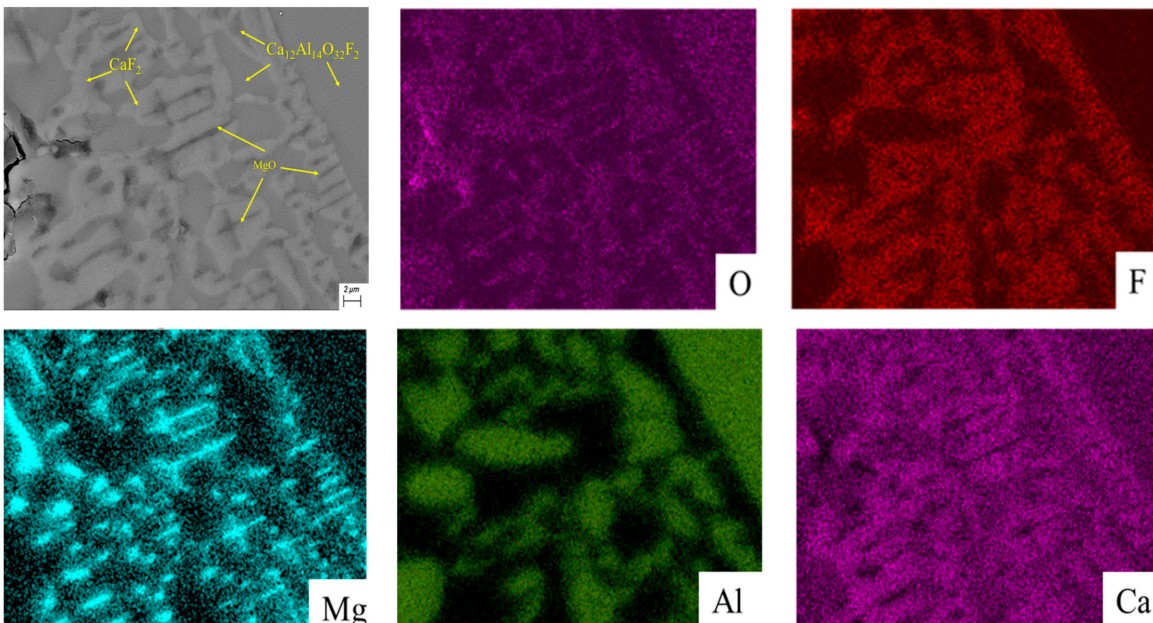

**Figure 7.** Elements mapping of the #12 slag quenched at the exothermic peak temperature named P3.

In Figures 6 and 7, the three crystalline phases are consistent with the XRD analysis. The CaF$_2$ crystal with dendritic and flow-like morphology was the dominant crystalline phase. The MgO crystal was thin with the stripe-like morphology or disperse with the faceted morphology.

The SEM-EDS image of the #13 slag quenched at the temperature of the exothermic peak named P3 is shown in Figure 8. The four crystalline phases are consistent with the XRD analysis. The MgAl$_2$O$_4$ was disperse with faceted morphology. The morphology of MgO is dendritic or stripe-like.

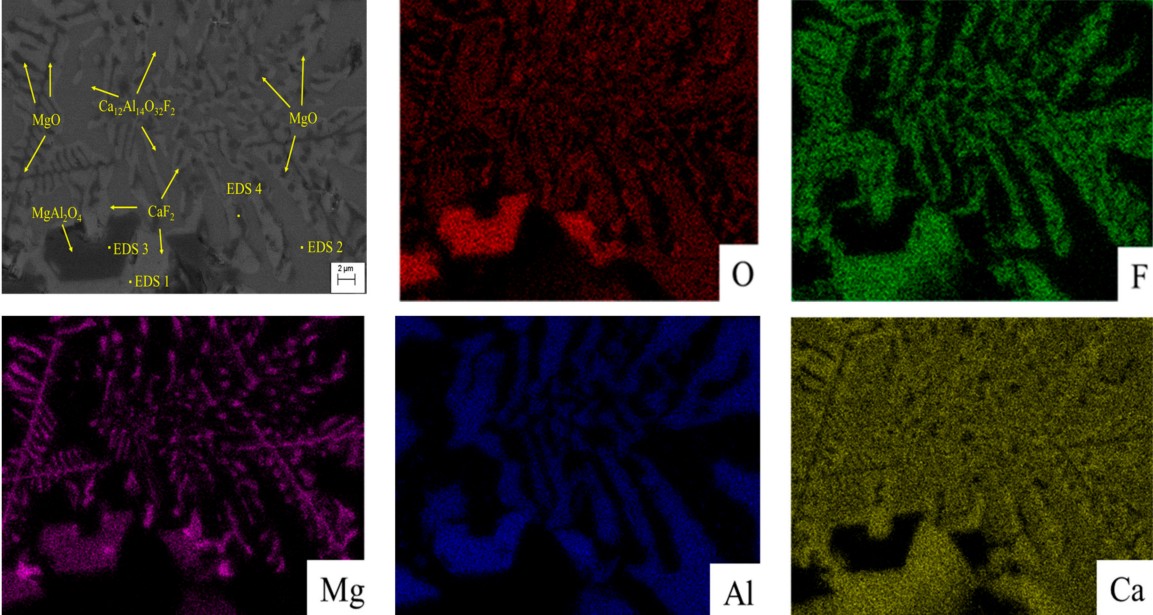

**Figure 8.** *Cont.*

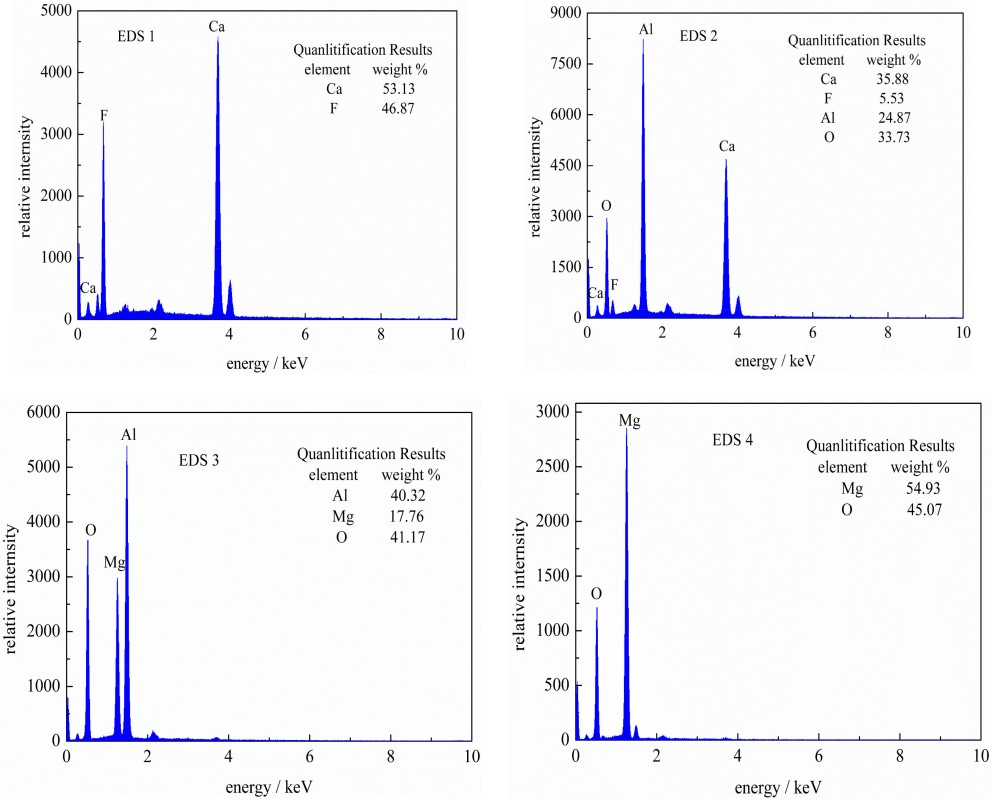

**Figure 8.** SEM-EDS results and elements mapping of the #13 slag quenched at the temperature of the exothermic peak named P3.

The SEM-EDS analysis of the #14 slag quenched at the temperature of the exothermic peak named P3 is shown in Figure 9. The crystalline phases analyzed by SEM-EDS were consistent with the XRD results.

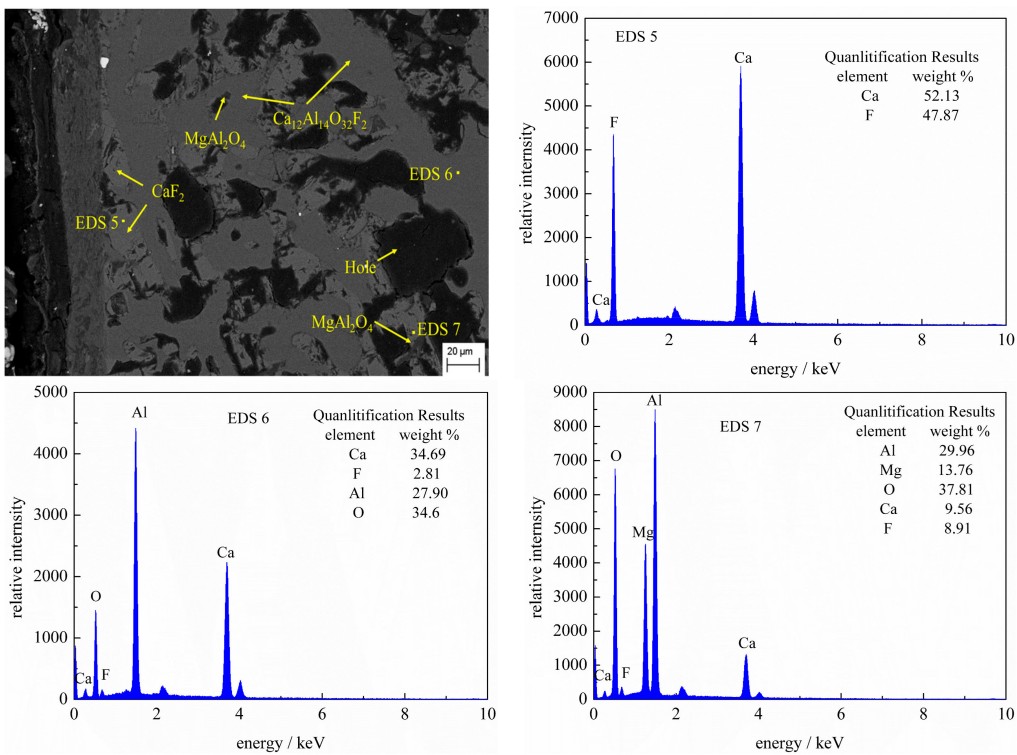

**Figure 9.** SEM-EDS results of the #14 slag quenched at the temperature of exothermic peak named P3.

The elements mappings of the #15 slag sample quenched at the temperature of the exothermic peak named P3 are shown in Figure 10.

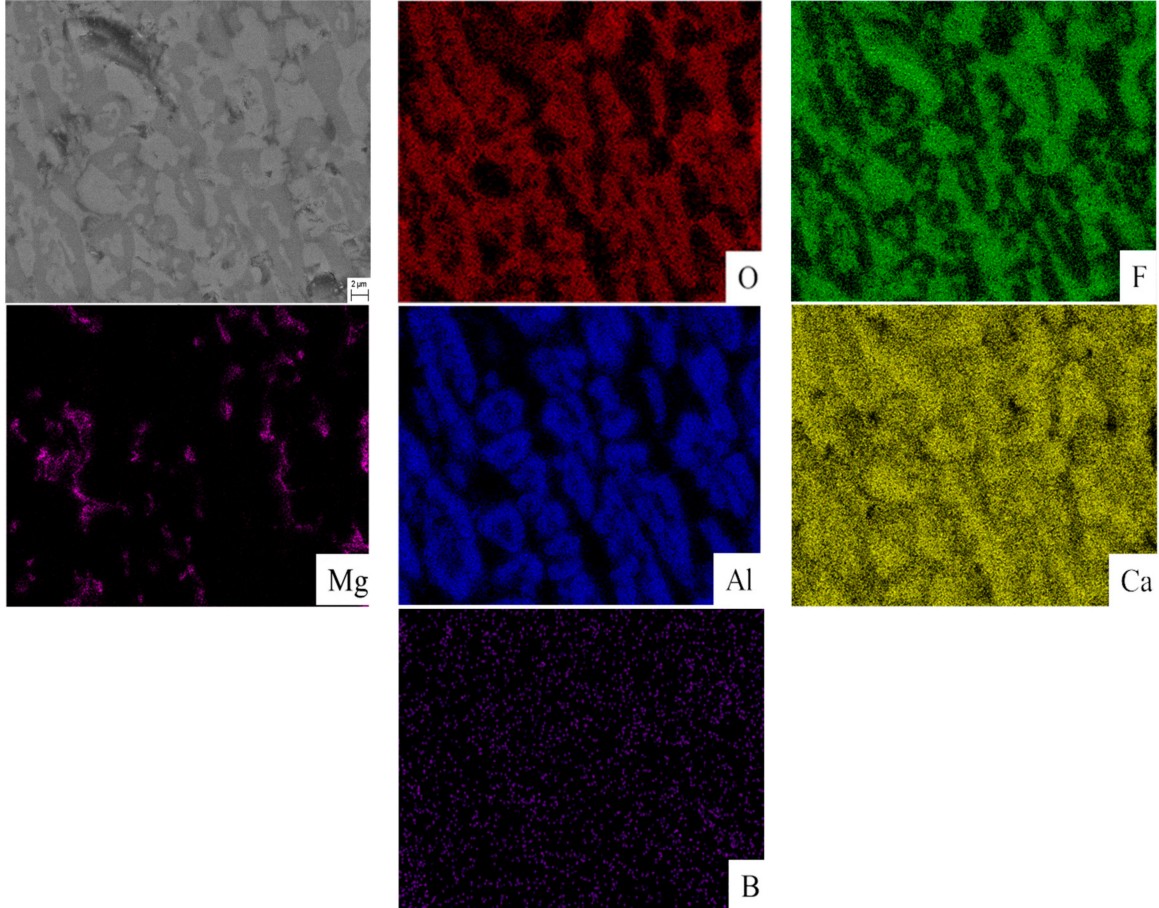

**Figure 10.** Elements mapping of the #15 slag quenched at the temperature of the exothermic peak named P3.

As illustrated in Figure 10, no clear enriched concentration of B was observed, although the crystalline phase of $Ca_3B_2O_6$ was identified by XRD analysis (results shown in Figure 3). The reason may be that at the $Ca_3B_2O_6$ precipitation temperature, the slag was viscous to inhibit the precipitation and growth of the $Ca_3B_2O_6$ crystal. The $Ca_3B_2O_6$ crystal size may be small, and its morphology could not be detected. Kashiwaya [30] also reported that the small size crystal was difficult to clarify by SEM-EDS analysis.

*3.4. The Calculated Results by Using FACTSAGE*

The solid fractions of the equilibrium crystalline phases of the #10–15 slags were calculated by using Factsage 7.2, and the results are shown in Figure 11.

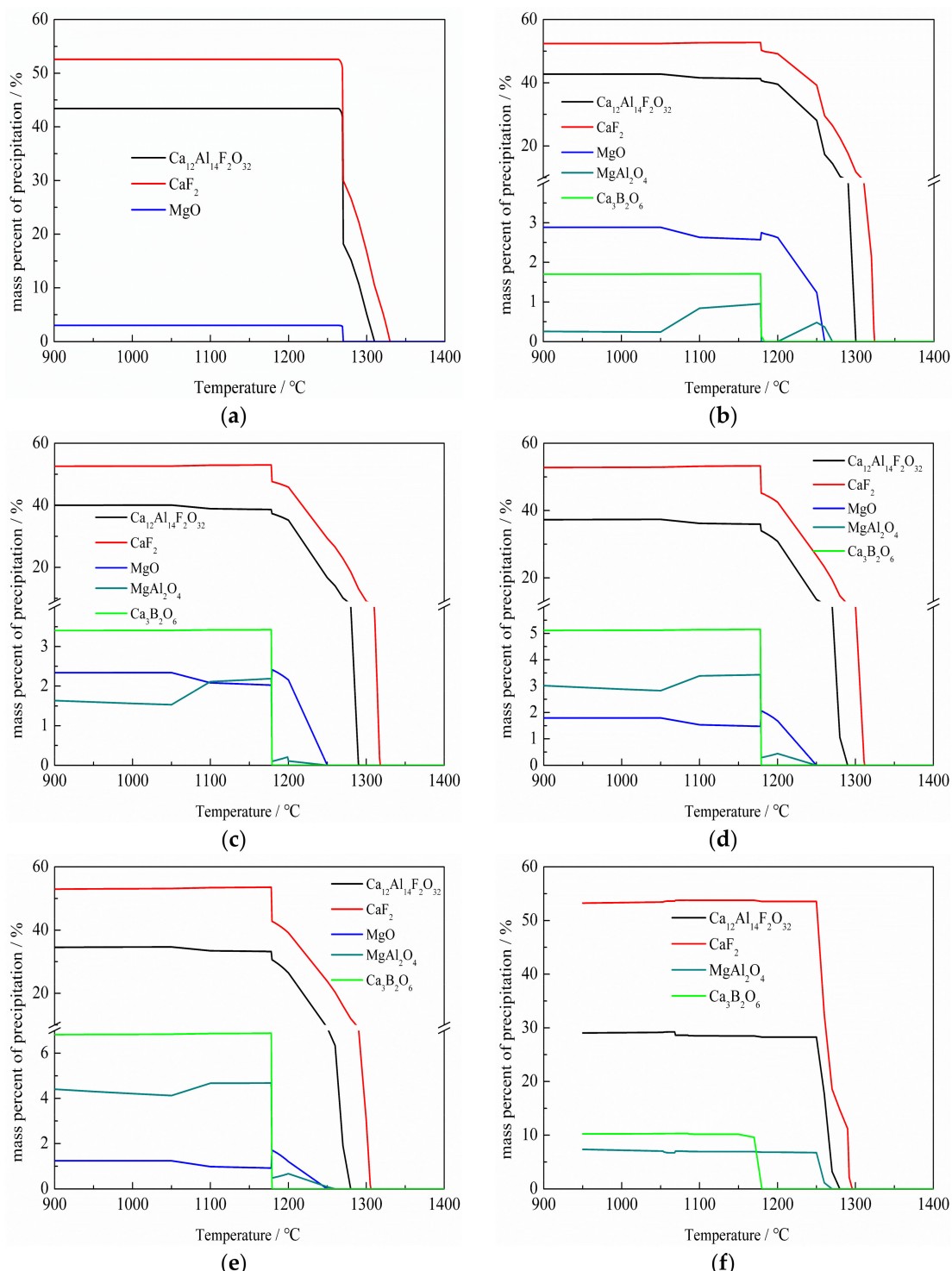

**Figure 11.** Mass of crystallization during cooling with temperature calculated using Factsage for the six slags, (**a**) #10 0.0% B$_2$O$_3$; (**b**) #11 0.5% B$_2$O$_3$; (**c**) #12 1.0% B$_2$O$_3$; (**d**) #13 1.5% B$_2$O$_3$; (**e**) #14 2.0% B$_2$O$_3$; (**f**) #15 3.0% B$_2$O$_3$.

In Figure 11, multiphases formed as the temperature decreased. The first crystalline phase that formed upon cooling for the six slags was CaF$_2$. The precipitation temperature of the first crystalline phase decreased with the increasing B$_2$O$_3$ content. The precipitation temperature of CaF$_2$ ranged from 1330 to 1290 °C. The precipitation temperature of Ca$_{12}$Al$_{14}$F$_2$O$_{32}$ decreased with the increasing B$_2$O$_3$ content, and ranged from 1307 to 1270 °C. The MgO and MgAl$_2$O$_4$ precipitated

almost simultaneously, and the precipitation temperature fluctuated between 1270 °C and 1250 °C. The precipitation temperature of $Ca_3B_2O_6$ decreased slightly from 1180 °C to 1170 °C as the $B_2O_3$ content increased. The total mass of the crystalline phase of $CaF_2$ was generally kept steady. The total mass of $Ca_{12}Al_{14}F_2O_{32}$ and MgO reduced, while the $MgAl_2O_4$ and $Ca_3B_2O_6$ increased with the increasing $B_2O_3$ addition. MgO did not precipitated in the #15 slag, and $MgAl_2O_4$ and $Ca_3B_2O_6$ did not precipitate in the #10 slag.

Some deviations existed between the XRD results and the Factsage calculation results. This was possibly due to the Factsage prediction only providing results in equilibrium, while in practice, the slag system was generally not in an equilibrium state [31], or the amount of the precipitated crystalline phase was too small to be detected by XRD. The equilibrium content of $Ca_3B_2O_6$ was obviously larger than the equilibrium content of $MgAl_2O_4$/MgO calculated by Factsage, but $MgAl_2O_4$/MgO was identified, while the $Ca_3B_2O_6$ was not identified by XRD analysis in the #12–14 slags. This was possibly because during the cooling process, the precipitation temperature of $Ca_3B_2O_6$ (around 980 °C) was much lower than the precipitation temperature of $MgAl_2O_4$/MgO (around 1230 °C), and the low precipitation temperature was not beneficial for the growth and the precipitation of $Ca_3B_2O_6$.

### 3.5. Activation Energy of Crystallization

Based on the above analysis, the kinetics analysis of the $CaF_2$ crystallization in the 10#–15# slags are shown in Figure 12.

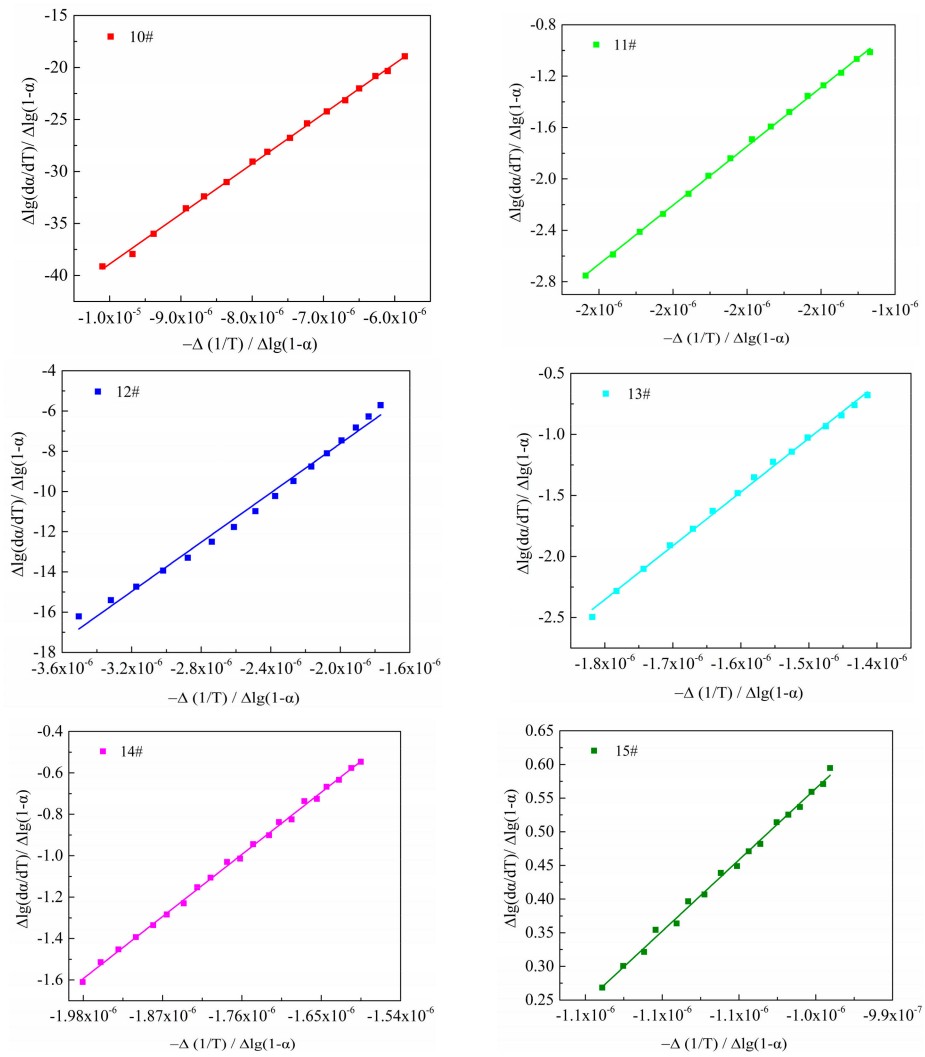

**Figure 12.** The kinetics analysis of the precipitate of $CaF_2$ in the #10–15 slags.

In Figure 12, the experimental data exhibited satisfactory linear relationships (the $R^2$ values for the six slags were 0.998, 0.997, 0.998, 0.997, 0.995, and 0.989, respectively). The yield values of the crystallization activation energy are shown in Figure 13. The Avrami exponent n of the #10–15 slags were all around 4 (4.1, 4.1, 4.3, 4.3, 3.8, and 4.0). Based on the morphology of $CaF_2$ (dendritic morphology) and the value of n, the crystallization behavior of $CaF_2$ was three-dimensional growth with a constant nucleation rate [32,33].

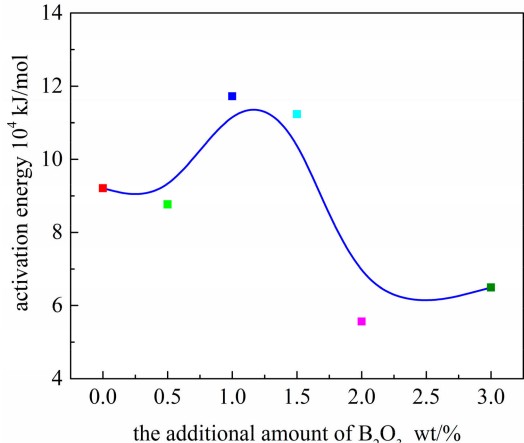

**Figure 13.** Calculated activation energy of the precipitation of $CaF_2$ with varied $B_2O_3$ contents.

The activation energy is an indication of the energy barrier that must be overcome for a crystal to form. Generally speaking, the lower the activation energy, the higher the tendency to crystallize. With the $B_2O_3$ content increasing, the crystallization tendency of the molten slag was inhibited ($B_2O_3$ addition ≤ 1.5%). When the $B_2O_3$ addition ≥2%, the $B_2O_3$ addition was beneficial for the crystallization.

The activation energy of the crystallization phase of $Ca_3B_2O_6$ precipitated in the #11–15 slags was also determined similar to the analysis of $CaF_2$, and the results are shown in Figure 14.

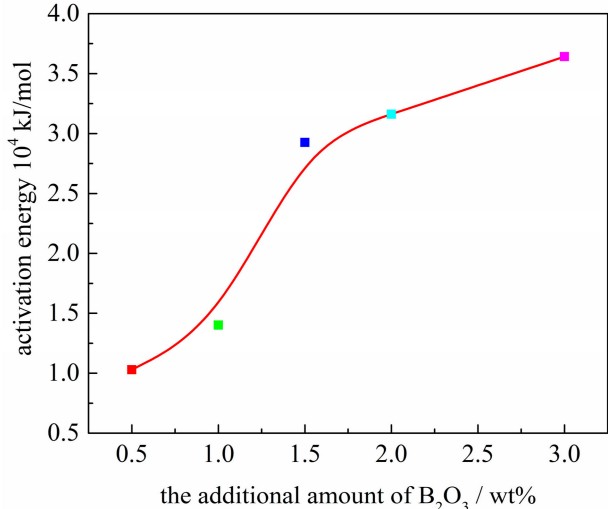

**Figure 14.** Calculated activation energy of the precipitation of $Ca_3B_2O_6$ with varied $B_2O_3$ contents.

In Figure 14, the activation energy of $Ca_3B_2O_6$ increased with the $B_2O_3$ addition increasing. The Avrami exponent n of the #11–15 slags were 1.7, 1.9, 3.8, 3.8, and 3.7. Since the morphology of the $Ca_3B_2O_6$ crystal was not able to be detected, the crystallization mechanisms of $Ca_3B_2O_6$ could not be determined.

The crystalline phase is easier to be formed between a solidified shell and copper mold in the slag melt with a stronger crystallization tendency, causing the formed slag film to be too thick during

the ESR process. This is unfavorable to provide appropriate horizontal heat transfer. Insufficient heat transfer is one of the main factors that should be responsible for surface defects and unstable ESR operation. Therefore, the additional amount of $B_2O_3$ should be proper. According to the above analysis, the additional amount of $B_2O_3$ should be around 1.0% to inhibit the precipitation of $CaF_2$ near the water-cooled mold.

## 4. Conclusions

The primary crystalline phase was $CaF_2$, and there was no change in the type of the primary crystalline phase with increasing $B_2O_3$ content at the cooling rate of 10 °C·min$^{-1}$, while the morphology of the $CaF_2$ crystal was mainly dendritic.

The sequence of crystal precipitation during the cooling process at the cooling rate of 10 K·min$^{-1}$ was $CaF_2$ to $Ca_{12}Al_{14}O_{32}F_2$ and $MgO/MgAl_2O_4$, followed by $Ca_3B_2O_6$.

The activation energy of the $CaF_2$ crystal increased at first; then, it decreased and reached stability with the increasing $B_2O_3$ content. Meanwhile, the activation energy of $Ca_3B_2O_6$ crystal increased with the increasing $B_2O_3$ content. The crystallization behavior of $CaF_2$ was three-dimensional growth with a constant nucleation rate.

To attain good surface quality, the metallurgical quality of the ingot, and stable ESR operation, the amount of $B_2O_3$ added into the $CaF_2$-based ESR slag should be around 1.0%.

**Author Contributions:** X.G. and Z.J. conceived and designed the experiments; L.P. performed the experiments; L.P. contributed to writing and editing of the manuscript. F.L. and H.L. provided the experimental resources.

**Funding:** This research was funded by the National Key R & D Program of China [Grant No. 2016YFB0300203] and the National Natural Science Foundation of China [51974076].

**Conflicts of Interest:** The authors declare no conflict of interest.

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
