# Peer review of "Effect of B2O3 on the Crystallization Behavior of CaF2-Based Slag for Electroslag Remelting 9CrMoCoB Steel"

_metals, doi:10.3390/met9121331_

Round 1

Reviewer 1 Report

Dear Authors,

I have read Your work with great attention and interests. I’m convinced that the manuscript falls within the scope of the journal and the work is sufficiently original, but I have a remarks.

I hope that they will be helpful in preparing the final version of the article.

I propose to add one keyword: CaF2.

Introduction chapter is well written. I propose do add short description of idea (principles) of electroslag remelting.

Line 33: From the context of the sentence and the role of B in iron alloys, it seems to me that the word "reinforced" should be replaced with "reinforcing".

Line 58: Please note the name of the FactSage software.

Lines 65, 95, 110, 111 e.t.c.: add space before unit

Figure 1: Descriptions are too small. Crucible is not indicated by a line (at least I don't see it).

Table 1 caption: Correct the typo in "the": it should be uppercase.

Lines 96, 100, 114 e.t.c. : correct unit

Lines 104 and 105: check grammar correctness.

Lines 124 and 127: There is no need to repeat the information on the use of the program in two sentences in a row.

Line 136: add reference for equation.

Lines 148-150: remove sentence from MDPI template.

Line 185: change “The” to “the”

Figure 8: Descriptions are too small.

Lines 263 and 264: this sentence should be given earlier, before Figure 10.

Figure 11 caption: change “mass” to „Mass”

Line 297/300: remove spaces from: “MgAl2O4 / MgO”

Figure 12 caption: change “the” to „The”

Line 312: “satisfactory linear relationships” should be confirmed by providing R2 coefficients values.

Figure 13 caption: add space before “CaF2

Figure 14: There is no need to enter values on the Y axis to the nearest hundredths.

References:

In my opinion, it is worth citing recent articles from MDPI publishing house journals (Metals, Materials, Applied Sciences, e.g.). This increases the visibility and impact of the article.

Reviewer 2 Report

The paper discusses about the effect of B2O3 on the crystallization behaviour of CaF2 based slag for electroslag remelting 9CrMoCoB steel. The paper has been fairly-organized and can be considered for publication if the following comments are considered by authors.

Please explain how the increase of B2O3 addition affects the exothermic peak. This phenomenon has been shown in Fig. 2 and needs more elaboration. I would recommend that authors talk about the texture and microstructure evolution due to the influence of B2O3. Therefore, please read and cite the following articles in your manuscript.

[1] Microstructure evolution and mechanical behavior of a new microalloyed high Mn austenitic steel during compressive deformation, Materials Science and Engineering A, 615 (2014) 424-435.

[2] Texture development of ARB-processed steel-based nanocomposite, Materials Engineering and Performance, 23 (2014) 4436-4445.

How did the authors determine that a special exothermic peak refers to the formation of combination of elements? For example, it has been implied that "the second exothermic peak represents the formation of Ca12Al14O32F2 and MgO in the 10#~12# slags". Based on Fig. 5, an EDS image as shown in Figure 5, how did the authors understand that the CaF2 crystal was the dominant crystalline phase and occupied the largest crystalline fraction? As shown in Figs. 12 and 13, please briefly explain how the activation energy for a precipitate is calculated? And how the addition of B2O3 affects it. The important question is that why there is no change in the type of the primary crystalline phase with increasing B2O3 content at the cooling rate of 10℃.min-1? Please also take a careful look and revise the quality of the English grammar and syntax where needed.

Round 2

Reviewer 2 Report

The authors tried their best to revise the manuscript based on my comments. The revisions are satisfactory. The paper can be considered for publication in the current format.